



# Low contributions of dimethyl sulfide (DMS) chemistry to atmospheric aerosols over the high Arctic Ocean

Miming Zhang[1,2], Jinpei Yan[*1,2], Qi Lin[1,2], Hongguo Zheng[3], Keyhong Park[4], Shuhui Zhao[1,2], Suqing Xu[1,2], Meina Ruan[1,2], Shanshan Wang[1,2], Xinlin Zhong[3], Suli Zhao[3]

[1] Key Laboratory of Global Change and Marine Atmospheric Chemistry, MNR, Xiamen 361005, China;
[2] Third Institute of Oceanography, Ministry of Natural Resources, Xiamen 361005, China;
[3] Thermo Fisher Scientific Co. Ltd., Shanghai, 610000, China.
[4] Korea Polar Research Institute, 26 Songdomirae-ro, Yeonsu-gu, Incheon 21990, Republic of Korea;

*Correspondence to*: Jinpei Yan (jpyan@tio.org.cn)

**Abstract.** The Arctic Ocean is continuously warming, resulting in sea ice retreat, which significantly impacts the marine biogenic sulfur cycle. The formation of aerosols from DMS oxidation and their climatic effects in the polar regions are of great concern. However, the impact of DMS chemistry on atmospheric aerosols in the high Arctic Ocean (AO) is still unclear due to the limitation of field observations and datasets. Gaseous methanesulfonic acid (MSA) and aerosol chemical species (MSA,

$SO_4^{2-}$ and DMA, etc.) were determined simultaneously with a high time resolution (1 hour) in the AO and Pacific Ocean (PO) to reveal the DMS chemistry in these regions. Extremely low MSA concentrations were observed in the high AO (75°-85°N), with an average of only $7.42 \pm 6.60$ ng•m$^{-3}$. However, high MSA concentrations, with an average of $168.60 \pm 167.60$ ng•m$^{-3}$ were observed in the mid-latitude regions (45°-60°N). Sea salt aerosols were the most dominant source in the high Arctic Ocean, accounting for 88.78% of the total aerosols, which was much larger than the values in the other regions. The MSA

fraction was much lower in the high latitude regions than in the other regions, accounting for only 1.61% of the total aerosol particles. The latitudinal distribution of MSA was consistent with that of DMS over the AO. Low DMS chemistry was determined based on the low DMS emissions in the HL region. These results highlight the contribution of DMS chemistry to atmospheric aerosols and extend the knowledge of how biogenic aerosols impact the regional atmosphere in the high AO.

**Short summary**

Extremely low contribution of DMS chemistry to the aerosols over the high AO was determined by the inhibition of marine phytoplankton, which extends the knowledge how will biogenic sulfur cycle impact the regional climate as AO sea ice retreat in the future



## 1.0 Introduction

The Arctic is known for its amplified of global climate change, which is warming faster than other regions of the world
(ACIA, 2005; Dall' Osto et al., 2017). Over the past decade, Arctic warming has been accelerated, resulting in the reduction
of multiyear and single-year sea ice in the high Arctic regions (Gilgen et al., 2018; Vaughan et al., 2013). Climate models
predict that the Arctic will become ice-free in the late summer times in the next several decades, due to the continuous
temperature increases in the future (Browse et al., 2014). Sea ice acts as a barrier to the exchange of heat and materials between
the sea and air (Gilgen et al., 2018; Yan et al., 2020a). The emissions of marine aerosols increase, when sea ice retreat occurs
in the Arctic (Struthers et al., 2013; Deshpande & Kambra, 2014) and Antarctic regions (Yan et al., 2020a), as more sea salt
and biogenic gases (such as DMS, which is a precursor of secondary aerosols) are released into the atmosphere (Browse et al.,
2014). In addition, sea ice retreat may also increase the burden of anthropogenic aerosols in the Arctic Ocean (AO), because
more ships pass through the AO in the summer seasons due to the loss of sea ice (Gilgen et al., 2018).

Both anthropogenic and marine sources have significant impacts on the atmospheric aerosols in the AO. Particles transported
from the mid-latitude regions in Europe, North America and Asia (Fisher et al., 2011; Hegg et al., 2010), increase the aerosol
burden and change the regional climate in the Arctic (Xie et al., 2018). Such as black carbon can absorb solar radiation,
resulting in an increase in sea ice melting when it is deposited on the surfaces of snow and ice (Doherty et al., 2013; Flanner
2013). The relationship between aerosols and sea ice melting in the Arctic has been reported in previous studies (Dall' Osto et
al., 2017; Gilgen et al., 2018; Browse et al., 2014). Sea salt aerosol (SSA) concentration will increase by a factor of 2 - 3 in
the summer time in the Arctic by 2100 due to sea ice melting (Struthers et al., 2013). High fraction of Arctic aerosols is
composed of SSAs in the summer (Deshpande & Kambra, 2014).

The enhancement of biogenic gases emission in the summertime also increases the amounts of atmospheric aerosol in the
AO (Sharma et al., 2012). The loss of sea ice in the AO promotes the air-sea exchanges and subsequently increases dimethyl
sulfide (DMS) emissions. The oxidation of DMS to form methanesulfonic acid (MSA) and $SO_4^{2-}$ aerosols can modify aerosol
properties and participate in new particle formation in the marine atmosphere (DMS chemistry), increasing the aerosol
population in the marine atmosphere (Chang et al., 2011; Ghahremaninezhad et al., 2016), and subsequently changing the
climate directly by reflection solar radiation and indirectly by acting as cloud condensation nuclei (CCN) (Charlson et al.,
1987). The observations carried out in the Ny-Ålesund revealed that the contribution of biogenic DMS to the formation of
aerosol particles was substantial during the Arctic phytoplankton bloom period (Park et al., 2017). Chang (2011) found that
marine biogenic DMS has an important contribution to the NPF in the summertime in the Canadian Arctic and may possibly
increase the aerosol concentration due to the loss of sea ice in the coastal Canadian Arctic. Observations at Alter, Nunavut and
model simulation results also indicate that DMS controls the clean summertime Arctic aerosols and clouds (Leaith et al., 2013).
Arctic atmosphere becomes relatively free of anthropogenic aerosols due to the increase of wet deposition in the summertime
in the Arctic (Crof et al., 2016), but high MSA concentrations have been observed at various locations north of 70º (Sharma et
al., 2012; Willis et al., 2016). The response of marine DMS emissions to Arctic aerosols has been investigated in the coastal



Arctic and low Arctic (Leaith et al., 2013; Hayashida et al., 2017). However, there is still short of knowledge about how marine DMS emissions and other sources impact the atmospheric aerosols in the high AO, as direct observation datasets for this region are extremely rare. In recent years, as Arctic warming has become increasingly critical, the sea ice has retreated significantly during the summertime in the high Arctic, which makes it an excellent case to find out the linkage between marine DMS
emissions and Arctic aerosols in the high AO.

In this study, we aimed to quantify the contributions of marine DMS chemistry to the atmospheric aerosols over the high AO. The characteristics of aerosol sources over the high AO and (Pacific Ocean) PO were also analyzed to understand the changes in the natural and anthropogenic aerosol sources in these regions. The influence of DMS chemistry on the aerosols was also investigated in the high AO and low latitude regions to reveal how DMS emissions impact the atmospheric aerosols
over the AO, especially over the high Arctic.

## 2.0 EXPERIMENTAL METHODS AND OBSERVATION REGIONS

### 2.1 Observation regions

The observations were carried out on-board the R/V "Xuelong", which sailed from Shanghai, China, on 20 July 2018 and arrived in the high AO on 20 August 2018 during leg I, covering a large region of the PO and AO, (30°N to 85°N, 120°E to
140°W) (Fig. S1). Leg II started on 21 August 2018 in the high AO (84.8ºN, 166.75ºW) and ended on the 19 September 2018 in the PO (45.79ºN, 143.82ºE). Parts of leg II were in accord with parts of leg I.

### 2.2 Observation instruments and samplings

An ambient Ion Monitor-Ion Chromatograph (AIM-IC, URG9000D, Thermo Fisher Scientific Co. Ltd ) was used to analyze the water-soluble ion species and chemical compositions of the aerosols and gases, respectively (Fig. S2). The sampling inlet
connected to the monitoring instrument was fixed to a mast 20 meters above the sea surface. A total suspended particulate (TSP) sampler was positioned at the top of the mast. Conductive silicon tubing with an inner diameter of 1.0 cm was used as the connection to the instrument.

### 2.3 Measurement of water-soluble ion species

The measurement of gases and aerosol species using an onboard AIM-IC system has been described in detail by Yu et al.,
(2020). The sampling gas was firstly induced into a diffusion-based parallel-plate wet denuder (PPWD), in which the absorbable gases were diffused through two cellulose membranes and dissolved into the carrier stream (ultrapure water). Then, the carrier stream was corrected using a syringe pump for further IC analysis (Malaguti et al., 2015; Markovic et al., 2012). The particles were enlarged by the condensational growth in a particle supersaturation chamber (PSSC), where the particles were converted to droplets and subsequently collected by a cyclone to generate sampling solution. The water samples were
then divided into two equal portions using a syringe pump for anion and cation analysis. The calibrations of the different ion



species are illustrated in the Table S1 and S2, in which the $R^2$ value was greater than 0.999. The detection limits for $MSA^-$, $SO_4^{2-}$, $NO_2^-$, $NO_3^-$, $DMA^+$, $NH_4^+$, $Na^+$ and $Mg^{2+}$ were 0.18, 0.05, 0.07, 0.10, 0.17, 0.12, 0.1 and 0.05 ng m$^{-3}$ (aqueous solution), respectively.

### 2.4 Metrological data, sea ice and chlorophyll-a satellite data

Meteorological parameters such as the temperature, humidity, wind speed, and direction were measured continuously using an automated meteorological station deployed on the top deck of the R/V "Xuelong".

Remote sensing data was also used to achieve the spatial and temporal distributions of the sea ice and chlorophyll-a concentrations in the high AO to check the influence of marine phytoplankton activity on the DMS chemistry in the AO. The remote sensing chlorophyll-a from MODIS-Aqua (http://oceancolor.gsfc.nasa.gov) with a spatial resolution of 4 km. We used

the sea ice concentration data from the daily 3.125-km AMSR2 datasets (Spreen et al., 2008) (available at https://seaice.uni-bremen.de).

### 3.0 RESULTS AND DISCUSSION

### 3.1 Spatial distributions of gaseous and particulate MSA in the Arctic Ocean

As a condensable product of DMS oxidation, MSA is a tracer for linking particles with DMS. The spatial distributions of

the particulate MSA (MSAp) and gaseous MSA (MSAg) are illustrated in Fig. 1. MSAp ranged from under detection to 692.4 ng•m$^{-3}$, with an average of 41.9 ± 90.4 ng•m$^{-3}$, along leg I (Fig. 1a and Table S3), while only low gaseous MSAg concentrations were present, with an average of 9.4±7.1 ng•m$^{-3}$. Leg I observations covered a large scale of latitudes from 30°N to 85°N. Four main regions were characterized, including low latitude (LL),mid-latitude (ML)、sub-high latitude (SL) and high latitude (HL) regions, seen in Table S3, to better understand the spatial distributions of MSA in the different sea regions. The highest

MSAp concentration occurred in the ML region, with an average of 168.6±167.6 ng•m$^{-3}$, followed by the LL region (average of 57.9±38.5 ng•m$^{-3}$). However, an extremely low MSAp concentration was observed in the HL region, ranging from 1.5 to 23.3 ng•m$^{-3}$ with an average of 6.0±6.4 ng•m$^{-3}$. In contrast to the MSAp, the highest MSAg levels occurred in the LL region, with an average of 21.0±12.3 ng•m$^{-3}$. The variations in the MSAg level were not always consistent with those of the MSAp along the cruise tracks, indicating that the formation mechanisms of MSAg and MSAp from oxidation of DMS are different.

Similar results were observed in the SO (Yan et al., 2019). As seen in Fig. S3, the MSAg concentration decreased with the temperature during LL-leg I, conforming the preferential formation of MSAg at high temperature.

Different from Leg I, the highest MSAp concentration occurred in the SL region, with an average of 68.3±44.2 ng•m$^{-3}$ during leg II, which was two times higher than the MSAp value measured during SL-legI. The variations in MSAp levels in the SL region during leg I and II were associated with the phytoplankton activity in these regions (Fig. S6). The leg II tracks were closer to the coast, comparing with the leg I tracks. High chlorophyll-a concentration was present in this region. Abundant

DMS was possibly released into the atmosphere in higher phytoplankton concentration regions (Tortell et al., 2011; Zhang et





al., 2015), which enhanced the production of MSA in the atmosphere. The MSAp concentrations were much lower during ML-leg II than during ML-leg I, but the MSAg concentration measured during the ML-leg II was consistent with the MSAg measured during the ML-leg I (13.9±15.2 ng•m⁻³ and 10.0±5.9 ng•m⁻³, respectively, Table S3). Note that observations were

carried out during late July in the ML-leg I region and during early September in the ML-leg II region. Higher chlorophyll-a concentrations were also found during ML-leg I (Fig. S6a and S6b). It was worthy to note that the MSAp concentration was much higher than the nss-SO$_4^{2-}$ concentration during ML-leg I, but the MSAp level was lower than nss-SO$_4^{2-}$ during ML-leg II, indicating that the high value of MSAp during ML-leg I may have also been affected by long-term transport.

Low MSAp concentrations were measured along HL-leg I and HL-leg II, with an average of 6.0±6.4 ng•m⁻³ and 13.4±7.2

ng•m⁻³, respectively. As seen in the Fig. S3, the observations carried out in the HL region lasted from August 2$^{nd}$ to September 3$^{rd}$, extremely low MSAp and nss-SO$_4^{2-}$ concentrations were observed in this region. It should be noted that MSAp and nss-SO$_4^{2-}$ concentrations changed scarcely during this period, suggesting that the DMS chemistry has little effect on the atmospheric aerosols in the high latitude of AO. Though obvious sea ice retreat occurred from July to September (Fig. S7), the chlorophyll-a remained at an extremely low level in this region. This phenomenon was very different from the observation

results in the high latitude SO (Yan et al., 2020a). As high DMS chemistry was reported in the high SO when the sea ice melted. However, increased phytoplankton levels did not occur in the HL when sea ice melted.

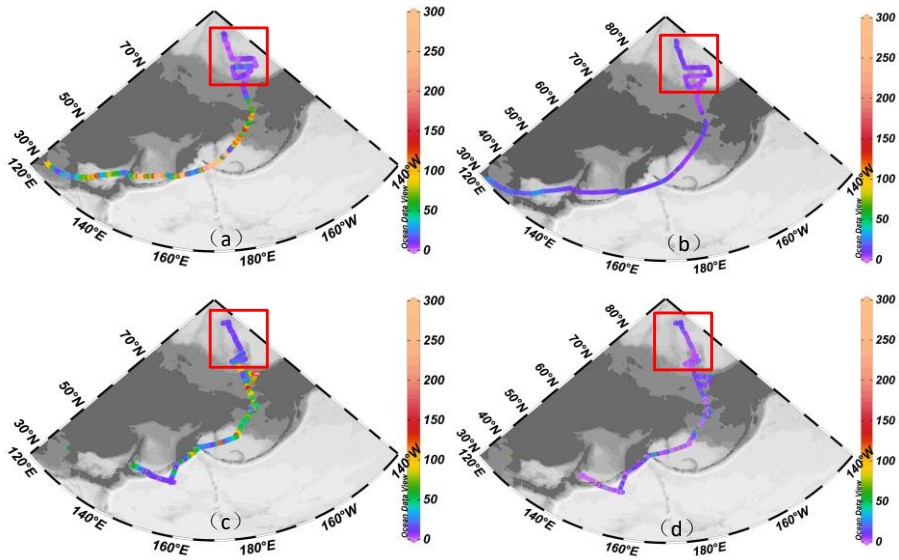

**Fig.1 Spatial distribution of the gaseous and particulate MSA concentrations, (a) Particulate MSA concentration during leg I**

**(ng.m⁻³); (b) Gaseous MSA concentration during leg I; (c) Particulate MSA concentration during leg II (ng.m⁻³) and (d) Gaseous MSA concentration during leg II.**





## 3.2 Characteristics of aerosol components in the PO and AO

In order to better understand the changes of aerosols in the different sea regions, the latitudinal distributions of the different
aerosol species are shown in Fig. 2. Similar latitudinal distributions of MSA and $Na^+$ were found, with the highest MSA and
Na concentrations occurring in the ML region (Fig. 2a and 2b). Generally, MSA is produced exclusively from the oxidation
of DMS (Sorooshian et al., 2007), making it a helpful marker of marine biogenic aerosols (Legrand & Pasteur 1998). Sodium
is mainly derived from SSAs (Teinila et al., 2014). SSAs are commonly associated with the wind speed (Nightingale et al.,
2000). Higher wind speeds were observed in the ML region (Fig. S3), which increased the air-sea DMS fluxes in this region,
resulting in high MSA levels in the atmosphere. The similar latitudinal distributions of MSA and Na also reveal that both the
$MSAp$ and $Na^+$ are from marine sources.

The distribution of $NO_3^-$ was very different from those of $MSAp$ and $Na^+$ during the observation period. The highest $NO_3^-$
concentrations were found in the LL region, followed by the ML and SL regions. Extremely low $NO_3^-$ concentrations were
observed in the HL region. $NO_3^-$ is generally regarded as a secondary aerosol species in the atmosphere, which was mainly
from anthropogenic sources (Mazzera et al., 2001; Yan et al., 2015). Though $NO_3^-$ can also be derived from natural marine
sources (Wolff et al., 1995), the $NO_3^-$ concentrations from natural sources in the marine atmosphere are extremely low. High
$NO_3^-$ concentrations were observed in the low latitude region, indicating that the aerosols in this region were significantly
impacted by the anthropogenic emissions. $NO_3^-$ levels decreased with increasing latitude, with the average $NO_3^-$ concentration
decreasing from 434 ng•m$^{-3}$ in the LL to 28.12 ng•m$^{-3}$ in the HL region (Fig. 2c). Low $NO_x^-$ concentrations (average of about
23 ng.m$^{-3}$) have also been reported in the high SO (Yan et al., 2020a). Similar to $NO_3^-$, the highest dimethylamine (DMA) level
was found in the LL region and it decreased sequentially from the ML, to SL and HL regions. The amines in the atmosphere
are derived from multiple sources, including anthropogenic sources and natural sources (Yao et al., 2016; Place et al., 2017).
High concentrations of DMA are often observed in coastal urban areas (Liu et al., 2017). High DMA concentrations in the LL
region confirm that atmospheric aerosols in this region are significantly impacted by anthropogenic pollution. As seen in Fig.
S1, the observation tracks in the LL region are between two continents. That is the reason why high $NO_3^-$ and DMA
concentrations were found in the LL region. As seen in Fig. 2, the lowest MSA, $Na^+$, $NO_3^-$ and DMA concentrations all
occurred in the HL region, but the mean MSA levels in the HL were much lower than those of the other species. Since $Na^+$
and MSA can be used as a marker for sea salt aerosols and biogenic sulfur aerosols, respectively. The variations in the MSA
to $Na^+$ ratio is useful to understand the contribution of biogenic sulfur species in the marine atmospheric aerosols. The MSA
to $Na^+$ ratios were very low in the HL, comparing to the values in the other regions (Fig. S4). It indicated that MSA contribution
from the DMS chemistry in the HL was much lower than those in the other regions.


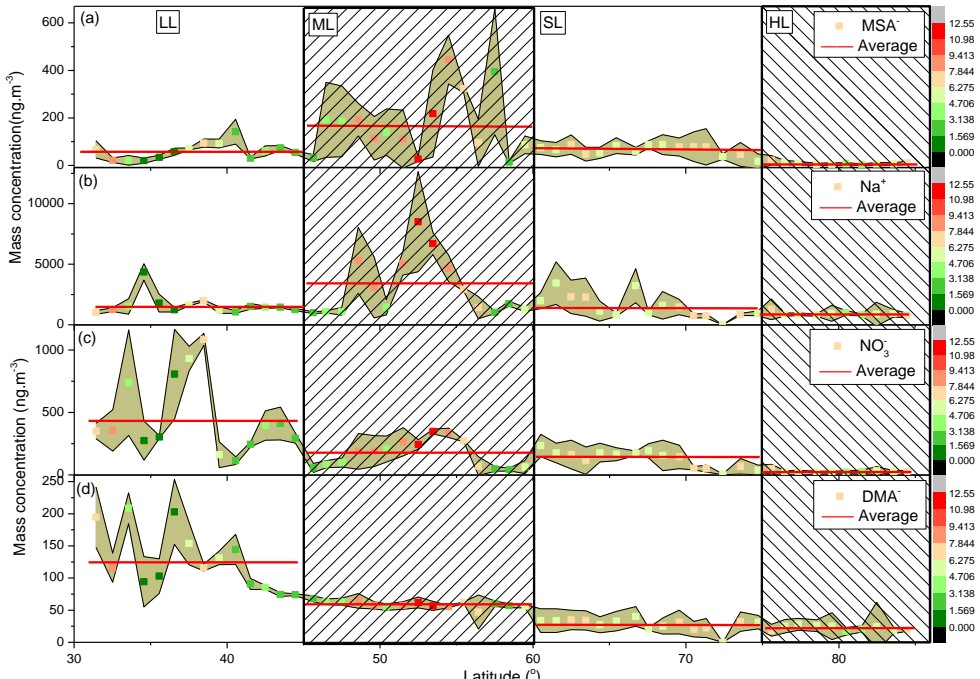

**Fig. 2 Latitudinal distributions of different aerosol species from 30-85ºN, (a) MSA concentration; (b) Na⁺ concentration; (c) NO₃⁻ concentration; and (d) DMA concentration.**


### 3.3 Contributions from different sources to the atmospheric aerosols in the AO

The atmospheric aerosols are significantly affected by the different sources, according to the latitudinal distributions. In this study, the aerosol particles were further classified into five types, including biogenic sulfur aerosols, sea salt aerosols, amines aerosols (MA, DMA, TMA, DEA and TEA), $NO_x^-$ aerosols ($NO_2^-$ and $NO_3^-$) and other aerosols. Sea salt, biogenic sulfur, 

amines and $NO_x^-$ were the major aerosol components in the marine atmosphere, accounting for more than 95% of the total aerosols. The other ion species (such as formic acid, acetic acid, and succinic acid etc.) only accounted for less than 5%, hence, we did not discuss them in this study.

Sea salt aerosols were the dominant sources during the observation cruise, accounting for more than 70% of the total aerosols (Fig.3). SSAs are generated from sea bubble-bursting, which is mainly determined by the wind speed. $Na^+$, $Mg^{2+}$, $K^+$, $Cl^-$, $F^-$, 

and $Br^-$ are considered to be the SSA components, and are commonly found in the sea water. Significant correlations between these species were observed in the previous study (Yan et al., 2020a). SSAs fraction accounted for more than 88.78% of the total aerosols in the HL region, followed by 80.81% in the ML region, 76.00% in the SL region and 71.09% in the LL region. High faction of sea salt aerosols (85%) was also observed in the open water around Antarctica (Dall' Osto et al., 2017). SSAs are strongly associated with the wind speed (Fig. S3). The lowest $Na^+$ concentration was observed in the HL region, comparing





with the Na⁺ concentration in the other regions. As seen in Fig. S7, a high concentration of sea ice was present in the HL region, which reduced the exchanges between air and sea, resulting in a decrease in SSAs emissions. Sea ice acts as a barrier to air-sea interactions and depressed SSA emissions have been reported in the SO (Yan et al., 2020a). Low Na⁺ concentrations in the HL region confirm the decrease in SSAs emissions in high sea ice concentration regions. However, the highest SSAs fraction occurred in the HL region, suggesting that the contributions from other sources were limited in this region.

Biogenic sulfur aerosols comprising $MSA^-$ and $nss\text{-}SO_4^{2-}$ are generated from the oxidation of DMS. High fraction of biogenic sulfur aerosols was observed in the ML region, accounting for 10.67%, followed by 7.99% in the SL and 6.28% in the LL region. However, the biogenic sulfur aerosol fraction was much lower than in the other regions, accounting for only 1.61% of the total aerosols in the HL region. DMS and MSA observations in Alert, Nunavut have shown that DMS emissions control the clean summertime Arctic aerosol and cloud (Leaith et al., 2013), but this may only occur in the coastal areas of the

AO. Such as, high MSA concentrations were observed at the Ny-Ålesund, Svalbard (78.5°N, 11.8°E), during Arctic phytoplankton blooms (Park et al., 2017). However, in the central high AO, the atmospheric aerosols were dominated by the sea salt particles. The DMS chemistry has little effect on the atmospheric aerosols. This phenomena is different from the observation results in the SO, where high biogenic sulfur aerosols have often been observed in the high sea ice coverage regions in the SO (Yan et al., 2020a). The biogenic sulfur aerosols in different sea regions are tightly associated with

chlorophyll-a (Fig. S6), indicating that the biogenic sulfur species are determined by the DMS emissions from marine phytoplankton activity.

    $NO_x^-$ aerosols, including $NO_2^-$ and $NO_3^-$, are secondary aerosols in the atmosphere, which were mainly derived from anthropogenic sources (Xu et al., 2013). High $NO_x^-$ fractions were observed in the LL and SL regions, accounting for 13.75% and 7.16%, respectively. As seen in the Fig. S1, the cruise tracks in the LL and SL regions were closer to the continent. That

is the reason why the high $NO_x^-$ fractions occurred in the LL and SL regions, as atmospheric aerosols are more easily affected by anthropogenic sources in areas closer to the continental sources. Similar to $NO_x^-$, high fractions of amine aerosols were found in the SL and LL regions. But the amine fraction (5.23%) in the SL region was higher than that (4.2%) in the LL region, because the amines emissions from marine surfaces. As higher chlorophyll-a levels were found in this region (Fig. S6). Good correlation between amines and chlorophyll-a has been observed in the coastal and marine areas in previous studies (Zhang et

al., 2013). In the low latitude regions, the amines are mainly affected by the anthropogenic sources, however, the amines were determined by the marine sources in the high latitudes. Low amines fraction was also found in the HL region, which is consistent with the biogenic sulfur aerosol fraction. This confirms that the low DMS chemistry was caused by the low phytoplankton activity in this region.





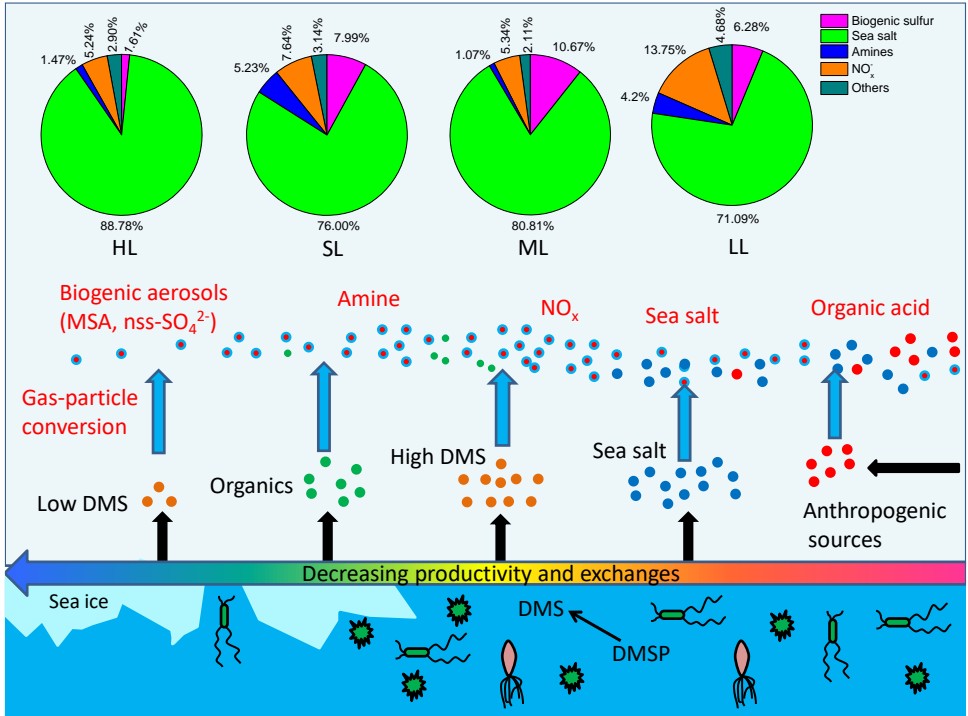

Fig. 3. Contributions of different sources to the atmospheric aerosols in the PO and AO

## 3.4 Determinant of low DMS chemistry over the high Arctic Ocean

MSA and nss-$SO_4^{2-}$ are the main products of DMS chemistry in the atmosphere (Zhang et al., 2013; Barnes et al., 2006). Generally, nss-$SO_4^{2-}$ is mainly associated with biogenic sources in the high latitude sea areas (Read et al., 2008), but nss-$SO_4^{2-}$ is also impacted by anthropogenic sources and volcanic sources (Hayashida et al., 2017; Legrand et al., 1998). A low $NO_x^-$ fraction was also found in the HL region, indicating that the contribution of the anthropogenic sources was low in this region. Low MSA and nss-$SO_4^{2-}$ concentrations were present in the HL region (Fig. S3). Nss-$SO_4^{2-}$ correlated well with MSA in the HL region, indicating that nss-$SO_4^{2-}$ is mainly derived from the DMS chemistry in the HL region. Therefore, MSA can be used as a marker to characterize the products from DMS oxidation in the atmosphere in this study. The oxidation of DMS with different radicals and its products in the atmosphere have been studied (Zhang et al., 2013; Barnes et al., 2006). MSA is generated through alternative routes, including gas phase reaction and reactive uptake on the exiting particles (Willis et al., 2016). Temperature has an important impact on the products of DMS oxidation in the atmosphere, as low temperature was in favor of MSA formation in the atmosphere (Bates et al., 1992). Negative relationship between MSA concentration and temperature was present in the ML region (Fig. 4a), corroborating the impact of temperature on MSA formation. But the MSA levels did not always increase with the temperature decreasing, positive correlation between MSA and temperature was found in the SL region in this study. The impact of temperature on the MSA concentration was tiny in the HL and LL regions. Note





that the temperature ranged from -7°C to 2°C in the HL region, and it ranged from 15 °C to 28°C in the LL region, but the MSA levels barely changed with temperature in both regions (Fig. 4a). It indicates that the temperature was not the main factor determining the products from the DMS oxidation in the atmosphere. RH is another factor that affects the formation of MSA. A high value of RH benefited the MSA formation, as DMS oxidation commonly occurred with aqueous reactions in the marine atmosphere (Barnes et al., 2006; Hoffmann et al., 2016). The MSA concentrations exhibited an increasing trend in the ML and SL regions when the RH was higher than 90%, confirming that a high RH benefited the formation of MSA. However, we did not find an obvious relationship between the MSA concentration and RH in the HL region. The MSA levels changed little with the RH in this region (Fig. 4b), indicating that RH had little effect on the MSA formation in the HL region.

Generally, MSA⁻ concentration is mainly determined by the DMS, radicals and environmental conditions (solar radiation, temperature and relative humidity etc.) (Barnes et al., 2006). The DMS air-sea flux increases with the wind speed (Nightingale et al., 2000), which increases the DMS emissions from the sea surface, resulting in an increase in the MSA concentration in the atmosphere. However, the MSA concentrations did not reveal a positive correlation with wind speed in this study (Fig. 4c), especially in the HL regions, the MSA levels were almost independent with wind speed. Though the DMS fluxes increased with wind speed, the atmospheric DMS levels were strongly limited by the DMS levels in the sea surface.

The DMS concentration was not measured during this cruise. However, the sea surface DMS concentration was determined during August and September 2014 (the 6th Chinese Arctic Expedition Cruise) in a similar region (Zhang et al., 2021). The cruise tracks of the 6th Chinese Arctic Expedition Cruise were similar with the tracks in this study (9th Chinese Arctic Expedition cruise). High DMS concentration was observed in the SL region (seen in Fig. S5). DMS concentration decreased as the latitude increased. The sea surface DMS remained at extremely low levels when the latitude was higher than 74°N. The latitudinal distribution of the MSA was consistent with the DMS variations in this study. These results confirm that the low contribution of the high AO atmosphere was controlled by the low DMS concentrations in this region. It is interesting that a good positive correlation ($R^2$=0.84, slope=43.7) was found between MSA and DMS when the DMS concentration was less than 2 nmol•L⁻¹ (Fig. 4d). However, when the DMS concentration was higher than 2 nmol•L⁻¹, the MSA concentration did not increase continuously with the DMS concentration. The MSA concentration was almost independent of the DMS concentration, indicating that DMS levels were no longer the limiting factor of the MSA formation. Similar results were observed in the SO, MSA concentration was not associated with DMS level, when the DMS concentration was high (Yan et al., 2020b). In this case, the radicals concentration became the limitation factor of the MSA formation in the atmosphere. The relationship between MSA and DMS (Fig. 4d) demonstrated that DMS concentration was the determining factor of the DMS chemistry over the high AO. Low DMS concentration was the critical factor for MSA formation, but radicals would become the key parameter of the DMS chemistry when the DMS concentration was high. The observation results confirmed that the low MSA concentration was determined by the low DMS concentration in the high AO and further demonstrated that low contribution of DMS chemistry was determined by the low DMS emissions in this region.



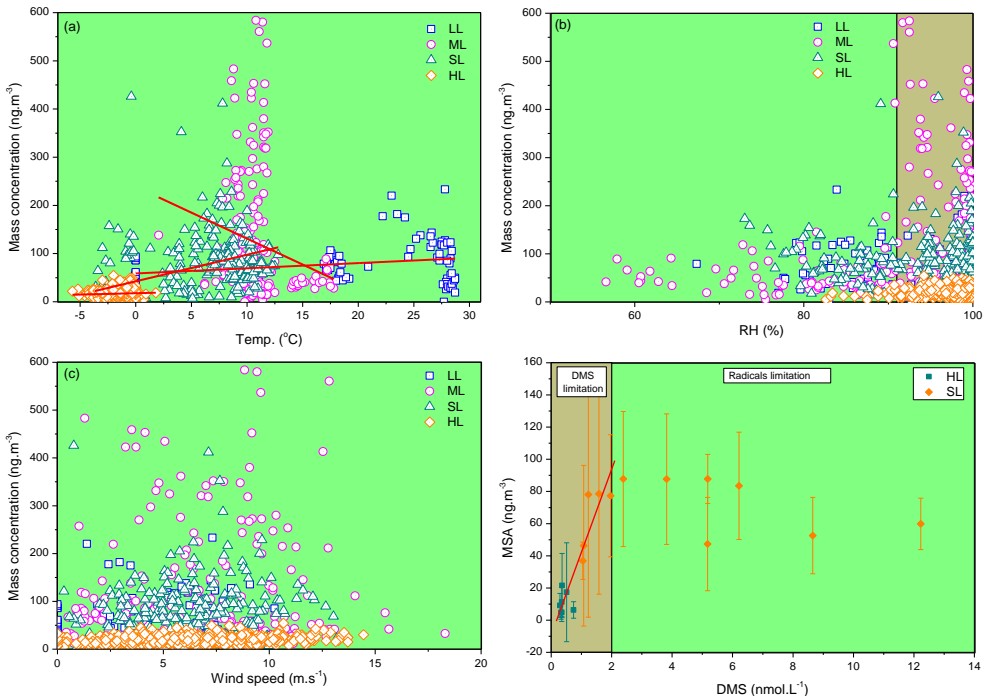


**Fig. 4 Influence factors on the MSA formation, (a) correlation between MSA and temperature, (b) Correlation between MSA and RH, (c) Correlation between MSA and wind speed, (d) Relationship between MSA and DMS.**

### 3.5 Implications of the low DMS chemistry in the high Arctic Ocean

DMS derived from the sea surface is the largest natural sulfur source in the atmosphere, constituting approximately 25% of the total global sulfur emissions into the atmosphere (Lana et al., 2011). DMS is easy to be oxidized to $SO_4^{2-}$ and MSA, modifying the aerosol properties. Sulfur acid is more effective to new particle formation (NPF), while MSA is more likely to grow on existing particles (Hayashida et al., 2017). The products from the DMS oxidation significantly impact the atmospheric aerosols and increase the CCN population in the marine boundary layer (MBL), resulting in changes in the radiation budget

and climate. As low aerosol burden was present in the high Arctic and Antarctic Ocean, the emission of DMS and its oxidation products highly impact the properties of atmospheric aerosols in these regions, especially when the DMS emissions increase evidently.

Biogenic aerosols produced from the DMS oxidation contribute more than 20% of the total aerosols in the high SO. However, biogenic aerosols derived from the oxidation of DMS only account for 1.61% of the total aerosols in the high AO. There,

different from the DMS chemistry in the SO, the DMS chemistry scarcely impacted the atmospheric aerosols in the high AO. Sea ice retreat in the Arctic and Antarctic Ocean is expected in the future because of continuous global warming. Previous studies have found that the sea ice retreat will increase biogenic aerosol emissions by more than 15% in the high SO. This



means that the changes of sea ice in the SO will significantly increase the biogenic aerosol burden in the SO, which will modulate the CCN populations and properties. The increase of CCN population enhances the reflection of solar radiation,
declines the sea surface temperature and affects marine phytoplankton activity and subsequently decreases the DMS production. Hence the DMS emissions and chemistry in the high SO are very important to the regional climates. But this situation would be very different in the high AO. The modeling simulations predict that the sea ice areas of AO will decrease from $6.1 \times 10^6$ to $3.4 \times 10^6$ km$^3$ in late summer by 2050 due to the rising temperature (Dall' Osto et al., 2017). Sea salt aerosols are the dominant source in the high AO, accounting for more than 88%. Thus, the sea ice retreat in the AO will significantly increase the sea
salt aerosol emissions. Sea salt aerosol emissions increased by more than 30% when the sea ice concentration decreased from 85% to 29% in the SO (Yan et al., 2020a). However, the sea ice melting in the high AO may not evidently increase the biogenic aerosols in these regions. As seen in the Fig. S6, the Chl-a levels were extremely low in this region even when the sea ice melted in the high AO. That means the phytoplankton activity was restricted by the limited nutrients in this area, resulting in the low DMS production and emissions (Zhang et al., 2021). As discussed in this study, the MSA formation is determined by
the DMS levels from the sea surfaces. It indicates that, the contribution of DMS oxidation products to the atmospheric aerosols is limited in this case. And the biogenic sulfur species from the marine phytoplankton activity may hardly affect the high AO atmospheric aerosols. However, the increase in sea salt aerosols due to sea ice retreat should be of concern in the high AO. Moreover, the sea ice melting in the AO will intensify the human activities in these regions, resulting in the increase of anthropogenic pollution emissions, especially the ship pollutant emissions, which will increase the aerosol burden and change
aerosol properties. Hence, anthropogenic emissions should also be highly concerned in the high AO in the future.

**Data availability**

The data used in the figures: the information, including wind speed and direction, temperature, RH along the cruise tracks were obtained from the shipboard metrological measurement system; The mass concentrations of water-soluble ion species in the particle and gas were obtained from the IGAC; Sea ice data was download from the website https://seaice.uni-bremen.de;
chlorophyll-a was obtained from MODIS-Aqua at http://oceancolor.gsfc.nasa.gov. Please also directly contact the author Jinpei Yan (jpyan@tio.org.cn) for the details.

**Supplement**

The supplement information was presented and available online.



**Author contribution**

JY conducted the observations, analyzed the results, and wrote the paper. QL and HZ contributed the data analyses and on-board technique assistance. MZ conducted the on-board observations and paper preparing. KP contributed to the refining the ideas and contributed considerably to the interpretation of the results. SX MR and SZ applied the calculations of sea ice distribution and Metrological data. SW,SZ and XZ contributed the observation data analyses. All authors were involved in discussing the results and improved the paper by proofreading.

**Competing interests:**

The authors declare no competing financial interest.

**Acknowledgements**

This study is Financially Supported by the National Natural Science Foundation of China (No. 41941014, 41305133), Scientific Research Foundation of Third Institute of Oceanography, MNR. (No. 2019024), Qingdao National Laboratory for

Marine Science and Technology (No. QNLM2016ORP0109), the Natural Science Foundation of Fujian Province, China (No. 2019J01120), the Chinese Projects for Investigations and Assessments of the Arctic and Antarctic (CHINARE2017-2020). The authors gratefully acknowledge Thermo Fisher Scientific Co. Ltd. China for the on board gas and aerosol monitoring system technical assistance and data analysis.

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
