# Peer review of "Low contributions of dimethyl sulfide (DMS) chemistry to atmospheric aerosols over the high Arctic Ocean"

_Atmospheric Chemistry and Physics, 2022_

## Referee Comment (RC2)

**Summary/recommendations**

The authors present gas-phase MSA and particulate MSA, DMA, SO4, etc. results from ship cruises taken in the northern Atlantic ocean. The results indicate that DMS chemistry may not greatly impact aerosol composition or concentrations in the study region. I think that it is a good look at the problem, but that there is still more research to be done before declaring this finding as fact. The authors did not collect gas-phase DMS, they were only out for a single season, and their study range was small. I think the paper should be published with more caveats and care to not over-state their findings. The paper was well cited and generally well written, but there are several sentence fragments throughout the paper. I have pointed out a few but I do encourage the authors to carefully go through the paper again.

**General comments**

Introduction: it would be helpful to define the different Arctic regions more rigorously- currently the authors refer to 'low', 'high', and 'coastal'. Please provide latitude ranges or similar.

As a general comment, it's helpful to define 'uncommon' acronyms in each major section when they first come up. (For example, AO and PO).

Methods: How did the authors quality control their datasets to remove ship emission influences (that is, emissions from the R/V Xuelong)? This is an important but missing detail.

Methods: there is no discussion of the size range of aerosol collected, which is relevant for climate and human health. Was there any differentiation by size?

Results: Since there was no direct measurement of DMS in the atmosphere, the findings are somewhat speculatory. The authors should state this limitation and be careful to not definitively state anything. The authors write as if they have solved this particular question for all of the AO (that DMS emissions and chemistry "scarcely impacted the atmospheric aerosols in the high AO" but I don't think this is so definitively solved. This study is a good look but requires more measurements/modeling over more seasons and a much larger area than the ship tracks covered. These caveats need to be discussed.

Lines 107-109: provide latitude ranges for LL, ML, SL, and HL. These ranges appear to be in Table S3 but I think this is relevant enough to be in the main text as well.

Lines 113-114: "The variations in the MSAg level were not always consistent with those of the MSAp along the cruise tracks, indicating that the formation mechanisms of MSAg and MSAp from oxidation of DMS are different." This is an interesting statement, I think the authors could

expand upon it some. Consider providing a supplemental figure or table showing the ratio of MSAg / MSAp. This may be of research interest to some.

Line 119-120: "The variations in MSAp levels in the SL region during leg I and II were associated with the phytoplankton activity in these regions (Fig. S6)." It is not easy to make this association based on looking at Fig S6 and Fig 1. The authors need to do more work here to make this association more clear. It would be nice to have some sort of statistical measure of how well associated MSAp and phytoplankton are - this could be an average for the region or more high-definition. The authors may also consider adding a supplemental figure that shows this association more clearly, like the ratio of phytoplankton to MSAp along the ship tracks, if possible.

Line 209-210: I recommend that the authors state "That is *likely* the reason why…" as the authors did not do a comprehensive analysis (for example, a principal components analysis) on where emissions were coming from for each species.

LIne 212: do the authors have the measurement precision to state that a  $\sim 1\%$  difference in amine is significant?

**Technical comments**

Line 22: HL not defined in the abstract

I recommend against using undefined acronyms in your short summary.

Line 29: "The Arctic is known for its amplified *rate*? of global climate change..." there is a missing word, perhaps the authors meant rate? Or Amplification instead of amplified?

Line 34: no comma needed between "increase, when"

Line 48 - I recommend adding a citation for "The loss of sea ice in the AO promotes the air-sea exchanges and subsequently increases dimethyl sulfide (DMS) emissions." - unless the Sharma et al 2012 citation was appropriate for this statement as well? This isn't clear to me.

Line 53: "The observations carried out in the Ny-Ålesund revealed..." Do the authors mean Ny-Alesund *region*?

Lines 98-99: the sentence here is a sentence fragment.

Results: the authors are inconsistent in whether or not they capitalize 'leg'.

Line 116: I think the authors meant "confirming" not "conforming"?

Line 135: the sentence here is a sentence fragment, I believe it belonged to the previous sentence?

Lines 167-168: the sentence here is a sentence fragment, I believe it belonged to the previous sentence?

Line 213: sentence fragment.

Line 258: suggest "strong positive..." rather than good. This is a more common way of phrasing it. (E.g. strong, moderate, weak correlations)

Line 277: Suggest "Sulfuric acid is more effective for new particle formation"

Line 305: The authors may have meant "concerning", not "concerned".

Citation for "Croft, B., Martin, R. V., Leaitch, W. R., Tunved, P., Breider, T. J., **D'Andrea**, S. D., and Pierce, J. R.: Processes controlling the annual cycle of Arctic aerosol number and size distributions, Atmos. Chem. Phys., 16, 3665–3682, doi:10.5194/acp-16-3665-2016, 2016. " - fix the D'Andrea, and this is erroneously "**Crof** et al 2016" in the main text, line 59. Fix to Croft.

**Figures/Tables**

All figure with spatial maps (e.g. Fig 1): I recommend considering explicitly drawing out the different characterized regions (ML, SL, HL, LL). For example, you could make the land/sea masses a lighter gray, then add dashed lines to separate each region + a label on the figure for each region. This would make the results more distinct.

Figure 2: it's not clear to me why ML and HL have dark cross hatchings over them? This makes the figure harder to read for me. I think the vertical lines are enough to provide distinctions. If the authors want to keep the cross hatchings, please make them a much lighter color, like light gray.

Table S3, Figure S3, Figure S4: define ML, SL, HL, LL in the table or figure caption.

---

## Author Comment (AC1)

**Response to reviewer 1**

**General response:**

**Dear anonymous reviewer 1,**

Thank you for your valuable comments on our manuscript "Low contributions of dimethyl sulfide (DMS) chemistry to atmospheric aerosols over the high Arctic Ocean". **We have carefully revised the manuscript as per the comments. Revisions in the text are shown using red highlight. The responses to the reviewer's comments** are marked in blue with the corresponding changes highlighted in red and presented in the following.

Zhang et al. measured ion concentrations in total suspended particulates (TSP) on a ship cruise from mid-latitudes to the Arctic during July-September, and focused on the analysis of MSA data. They found low MSA concentrations over the Arctic Ocean and concluded low contributions of dimethyl sulfide (DMS) chemistry to atmospheric aerosols over the high Arctic Ocean. The gas-phase and particle MSA dataset is useful to the community. However, the scientific justification throughout the manuscript is weak. The manuscript lacks novelty. The research method used is not supportive enough towards their conclusions. It has not reached the standard of publication at ACP, unless major revision is done.

Response: It is appreciated that the reviewer pointed out these weaknesses. However, it is very difficult for us to perform a perfect observation during the Arctic Ocean campaign, similar to a typical land station surrounding the Arctic Ocean. We can only obtain a short period of data across the central Arctic during our observation. To the best of our knowledge, up to now, few studies have reported the high-resolution observation particle or gas-phase MSA shipboard underway data over the Arctic Ocean ((Yu et al., 2021; Yu et al.,2020; Zhao et al., 2022), these publications are all from our group). We presented the details as follows to prove our conclusion of "low contributions of DMS chemistry to atmospheric aerosols over the high Arctic Ocean" was reliable and solid.

General comments

1. Atmospheric DMS was not measured in this study. The only seawater DMS data was from a

previous study in 2014. Therefore, a correlation between DMS and MSA cannot be reached. Both the emissions of DMS, the oxidation, and transport can affect the abundance of MSA observed during the cruise. The authors should not make strong claim of the contribution of DMS chemistry to atmospheric aerosols in the abstract and conclusion. They would not able to quantify the contributions of marine DMS chemistry to the atmospheric aerosols (Line 66).

Response:

Firstly, our previous study reported that deficient DMS levels (<0.5 nmol L$^{-1}$) and flux (general below 0.5 μmol m$^{-2}$ d$^{-1}$) were observed in the high latitude (Zhang et al., 2021, in Global Biogeochemical Cycles) due to the nutrient limitation and heavy sea ice cover. Even after sea ice retreat, the DMS levels remained unchanged, and the DMS flux slightly increased to only 1.2 μmol m$^{-2}$ d$^{-1}$. For the annual changes of DMS levels and flux, we can also conclude that these values were difficult to be changed as the upper layer water mass of Arctic Central was dominated by an increasing fresh water with very low nutrients (Figure 1, Zhang et al., 2021). Although we did not detect the atmospheric DMS and flux in the 2018 cruise, the low DMS flux and air DMS levels could be expected during the 2018 campaign.

[Figure]

Figure 1 Distribution of nutrients about surface seawater dimethylsufide (DMS) in the western Arctic Ocean. (a) Locations of conductivitytemperature-depth (CTD) stations along T1. Section R, panels (c)–(f), correspond to stations labeled R on the map in (a). (b). Relationships between DMS and Si and total N in surface water of all stations in (a). The panels on the right show depth profiles of (c) salinity, (d) Si, (e) total N, and (f) fluorescence

Secondly, we found that the MSA mass concentrations decreased from the low latitude Arctic to the high latitude. If the air mass contained high MSA levels could be rapidly transported to high latitude regions, we would observe high MSA in the high Arctic. However, we found deficient gas phase and particle MSA levels over there. In addition, MSA is well known only from DMS oxidation in the atmosphere. This means that the oceanic DMS emission greatly influences the atmospheric MSA production. Thus, the low emission of DMS in the high Arctic was possibly the main reason for extremely low MSA there.

Thirdly, our observation of aerosol compounds indicated that the contributions of biogenic sulfur decreased significantly (only 1.61%) in high latitude Arctic (Figure 4 in manuscript). This result suggested that the low DMS chemistry contribution to aerosol was found in the high Arctic.

2. The aerosols collected in this study is TSP. While MSA and sulfate are mostly in fine-mode aerosols, coarse-mode sea salt mass can make a large contribution to TSP. It is not surprising that they found a large contribution (88.78%) of sea salt aerosols to the total aerosols in the high Arctic Ocean on the ship 20m above the sea surface because coarse-mode sea salt could be important. The authors should clarify the potential contribution of coarse-mode sea salt to TSP and what impacts coarse-mode sea salt can have in the high Arctic Ocean. Please give more information on why we care about TSP over the Artic, instead of fine aerosols? We don't expect the coarse aerosols to go higher up and contribute to CCN etc.

Response: Thanks for the reviewer giving this good suggestion. Sure, the fine particle will make a significant contribution to the CCN. The sea spray might contribute largely to coarse-mode particles. However, in this study, we could only focus on the mass concentration in TSP for different kinds of ions because we did not measure chemistry characteristics in different particle sizes during the cruise. Thus, we can't reach any results or conclusions for the chemistry characteristic of fine or coarse-mode particles. The main purpose of this study is to analysis the characteristic of mass concentration of atmospheric ion distributions over different latitude oceans and their factors. We agreed that studying the fine particles and their chemicals would be more important, especially in different particle sizes. Thus, we have already purchased

instruments, like CCN Counter, Electrical Low-Pressure Impactor (ELPI), and Single Particle Aerosol Mass Spectrometer (SPAMS), since 2020. We anticipated to perform the investigations using all these instruments to solve the critical scientific questions pointed out by the reviewer in the Arctic Ocean in the future.

3. The fraction of MSA in TSP would be significantly affected by the coarse-mode sea salt mass, making it a less useful indicator of biogenic contribution to aerosol mass. Please clarify what we can learn from 1.61% of TSP (comprising fine and coarse aerosols) as MSA, as in the abstract.

Response: We agree that the fraction of MSA in TSP would be significantly affected by the coarse-mode sea salt mass. However, we disagreed that using the fraction of MSA in TSP is a less useful indicator of biogenic contribution to aerosol mass. In this study, although we only discuss the mass concentration of aerosol ions, either the mass concentration of MSA and DMA (an indicator of biogenic activity) indicated significant variations from the low latitude regions to the high Arctic Ocean or the fraction of biogenic sulfur aerosol decrease from low latitude region to the high Arctic. These results demonstrate that the contribution of mass concentration from biogenic sulfur aerosol to the TSP significance decrease from low latitude oceans to high Arctic Ocean.

4. More information is needed on how MSA is formed in the atmosphere.

Response: Sure, we added more information about how MSA is formed in the atmosphere in lines 58-62 as follows:

A major route of particulate MSA formation is the uptake of MSA on existing particles, and MSA uptake on different particles was different. Our previous studies indicated that sea salt particles are beneficial for MSA uptake (Yan et al., 2020 c). And, we also found great contribution of gas-phase MSA to total MSA, up to 31%, existing over the Southern Ocean atmosphere (Yan et al., 2019), and this result challenged the traditional understanding about gaseous MSA could be quickly converted to particulate MSA.

Other comments

1. Line 17: It is not clear whether the numbers for MSA concentration are for gaseous MSA

or particle MSA. It would be good to list both of them, make comparison, and explain the difference, since the gaseous and particle MSA data is the key in this manuscript.

Response: We revised the description and added more information in the Abstract as in lines 17-22 as follows.

The particulate MSA concentration indicated significant spatial variation with a decreasing tendency from the low latitude oceans to high AO. Extremely low particulate MSA concentrations were observed in the high AO (75º-85ºN), with an average of only $7.42 \pm 6.6$ ng•m$^{-3}$. In contrast, highest particulate MSA concentrations, with an average of $168.6 \pm 167.6$ ng•m$^{-3}$ were observed in the mid-latitude regions (45º-60ºN) in July. Generally, concentrations of gaseous MSA were much lower than those in particulate except in the area of high AO (with an average gaseous MSA value of around 8.4 ng •m$^{-3}$). In contrast, the highest gaseous MSA level was found in sub-high latitude (60 º-75ºN) with an average value of $24.2 \pm 46.8$ ng •m$^{-3}$.

2. Lines 19 and 20: Is it 88.78% of total suspended particle mass? 1.61% of total suspended particle mass?

Response: We revised the description in lines 23 and 25; we changed the total aerosol mass to "total suspended particle mass".

3. Line 22: HL is not defined.

Response: We added "high latitude (HL)" in line 27.

4. Line 41: Grammar issue with "such as".

Response: We revised it as "For instance, black carbon···" in line 45.

5. Introduction section: Since MSA is a focus of this study, there should be some introduction on how MSA is produced and lost in the atmosphere.

Response: In response to the comments above, we added some information about the formation of MSA particles as in lines 58-62 as follows:

A major route of particulate MSA formation is the uptake of MSA on existing particles, and MSA uptake on different particles was different. Our previous studies indicated that sea salt particles benefit MSA uptake (Yan et al., 2020 c). And, we also found great contribution of gas-phase MSA to total MSA, up to 31%, existed over the Southern Ocean atmosphere (Yan et al., 2019), and this result challenged the traditional understanding about gaseous MSA could be quickly converted to particulate MSA.

6. Section 2.2: Please provide more information about the sampling inlet setup (20 m above the sea surface) such as the flow and estimate the sampling lost of particles of different size during sampling.

Response: We provided more information about our sampling and measurement system in lines 90-93 as follows:

A total suspended particulate (TSP) sampler was positioned at the top of the mast to minimize the impact of ship emission. Conductive silicon tubing with an inner diameter of 1.0 cm was used to connect to the instrument to avoid the sampling loss of particles. The sampling flow of 16.7 L min$^{-1}$ was selected as the proper inlet flow for this system.

7. Section 2.3: It would be helpful to provide time resolution of the AIM-IC data. Is there any interference for the gas-phase MSA measurements, e.g. from MSIA?

Response: Almost no interference was found in the MSA measurement; the peak of MSA could be distinguished easily. The time resolution of this measurement method was 1 hour per sample along the cruise track. This sentence was added in lines 100-101.

8. Line 113-114: Please provide more information on how different are the formation mechanisms of MSAg and MSAp. Please also discuss the influence of gas-particle partitioning of MSA.

Response: We added more information to explain the formation mechanisms of MSAg and MSAp in lines 125-127 as follows:

Similar results were observed in the Southern Ocean (SO), and the reason could possibly be attributed to the temperature, in which high temperature (> 5°C) facilitated the formation of MSAg. In comparison, the condensation of MSAg onto particle surfaces was favorable at low temperature (< 5°C) (Yan et al., 2019).

9. Line 115-116: It is not that obvious form Fig. S3 that MSAg concentration decreased with temperature during LL-leg I. Please provide numbers or a scatter plot. In addition, does this negative correlation occur during LL-leg I or during the whole study? Why and why not? Does MSAp depend on temperature? Why and why not? What is the role of temperature-dependent gas-particle partitioning?

Response: It is difficult to find the straight relationship between temperature and MSAg. We can only make a general description that "the MSAg concentration generally decreased with the temperature during LL-leg I"; we can clearly see this variation tendency in Figure S3. And, for the relationship between temperature and MSAg, we did not want to spend too many sentences explaining it because this is not a new finding. We already discussed it in our previous work by Yan et al., 2019 "**Significant Underestimation of Gaseous Methanesulfonic Acid (MSA) over Southern Ocean**", published in Environmental Science & Technology. Please see the section "**Influence of Temperature on MSAg Levels in the SO.**" in this paper (Figure 2). Our main point is to explain why such low biogenic sulfur aerosols existed over the high AO atmosphere.

[Figure]

Figure 2 Influence of temperature on MSAg and MSAp. (a) and MSAp.(c) Effect of Correlation between MSAg and MSAp. (b) Impact of temperature on the correlation between MSAg temperature on MSAg to MSAT ratios.(Yan et al., 2019)

10. Line 115: Does SO refers to Southern Ocean? Please define.

    Response: Yes, it was revised as comments.

11. Line 128: It is not clear how to come to this conclusion "…indicating that the high value

of MSAp during ML-leg I may have also been affected by long-term transport…". Please clarify.

Response: We found the MSAp and nss-$SO_4^{2-}$ concentrations were different in these two periods. Commonly, the MSAp levels were found to be lower than nss-$SO_4^{2}$ caused by the DMS oxidation pathway and contribution of other sources of nss-$SO_4^{2-}$ (like human sources or long-range transportation). Generally, the ratio of MSA/ nss-$SO_4^{2-}$ was below 1. However, we found MSAp concentration was higher than nss-$SO_4^{2-}$. The unusual phenomena were possibly attributed to the air mass being influenced by the other source of high MSAp contained air mass. We revised the description as follows in lines 143-146:

Commonly, the MSAp levels were found lower than nss-$SO_4^{2-}$ levels, and the ratio of MSA/ nss-$SO_4^{2-}$ was below 1 (Zhang et al., 2015). However, our finding of a much higher MSAp levels than nss-$SO_4^{2-}$ here was possibly attributed to that the air mass was possibly influenced by the other air mass contained high MSAp through long-term transportation.

12. Line 131: It is not clear how to come to this conclusion "…It should be noted that MSAp and nss-SO42- concentrations changed scarcely during this period, suggesting that the DMS chemistry has little effect on the atmospheric aerosols in the high latitude of AO…". Please clarify.

Response: During our investigation period from August 2$^{nd}$ to September 3$^{rd}$ over the high AO, we found the concentration of MSA and nss-$SO_4^{2-}$ changed scarcely with very low levels. And, the MSA is only originated from DMS oxidation. According to our previous study (Zhang et al., 2021), the low DMS emission was investigated over the high AO before and after sea ice retreat. Thus, we concluded that DMS chemistry has a small contribution to the total suspended particle mass in the atmosphere. We revised the description to avoid misunderstanding as follows in lines 149-151.

It should be noted that MSAp and nss-$SO_4^{2-}$ concentrations changed scarcely during this period, suggesting that the DMS chemistry has a small contribution to the total suspended particle mass in the atmosphere of high AO.

13. Line 134-135: Grammar issue. Should be one sentence.

Response: We checked all these sentences. It was revised in lines 151-155 as follows:
Although obvious sea ice retreat occurred from July to September (Fig. S7), the chlorophylla remained at an extremely low level in this region possibly leading to very low DMS emission (Zhang et al., 2021, Figure S6). This would be the main reason for low observed sulfur aerosols. In contrast, our result differed from the observation in the high latitude SO that high atmospheric DMS chemistry contribution was reported when the sea ice retreated (Yan et al., 2020 a).

14. Lines 146 and 150: Na should be Na+. Please check throughout the manuscript.

   Response: It was revised, and the whole manuscript was checked.

15. Line 149-151: It is really difficult to see by eye that MSA and Na+ concentrations increased with wind speed. Please provide more information, such as numbers, scatter plots, etc.

   Response: Thanks for the reviewer pointing out this issue. However, we did not show that the MSA and Na+ concentration increased with wind speed according to our description in the manuscript. We wanted to present the high wind speed was found along the cruise track in ML region, and the high wind speed would increase the flux of DMS and result in high MSA concentrations.

16. Line 168: Grammar issue starting with "Since…".

   Response: We deleted "Since" here. The sentence was revised in line 186 as "Because $Na^+$ and MSA can be used as a marker for sea salt aerosols and biogenic sulfur aerosols, respectively, the variations in the MSA to $Na^+$ ratio is useful to understand the contribution of biogenic sulfur species in the marine atmospheric aerosols.".

17. Line 168-169: The authors state "…The variations in the MSA to Na+ ratio is useful to understand the contribution of biogenic sulfur species in the marine atmospheric aerosols…" Note that MSA is produced in fine aerosols while a large fraction of Na+ is present in coarse aerosols. The MSA/Na+ ratio in TSP may not be that useful to understand the contribution of biogenic sulfur species in the marine atmospheric aerosols. The information obtained from MSA/Na+ ratio in TSP could be significantly affected by freshly emitted coarse sea salt. Please discuss what that ratio stands for under this scenario. The aerosols in upper marine atmosphere would be different from the aerosols 20 m above the ocean surface.

   Response: We agreed that the aerosols in the upper marine atmosphere would be different from the aerosols 20 m above the ocean surface, and the information obtained from MSA/$Na^+$ ratio in TSP could be significantly affected by freshly emitted coarse sea salt. However, we only

obtained the data from 20 m height atmosphere; this data possibly indicated the status of atmospheric chemistry in the marine boundary layer (mixing layer). Additionally, our measurement could not distinguish the fraction of old or freshly sea salt and fine or coarse particles as well. The main purpose we concern about was to know the contribution of biogenic sulfur aerosol to the total suspended particle mass. We can only talk about the mass concentration contribution of each species.

18. Line 169-170: MSA/Na+ ratio doesn't lead to the conclusion of low MSA contribution from DMS chemistry in the HL. What is the role of Na+ here?

Response: We feel sorry that our understanding of the marine aerosols is different. Why the total suspended particle mass could not be used to discuss the contribution from biogenic sulfur aerosols. Even the coarse particle could be rapidly removed through dry or wet deposition; the fresh sea salt coarse particle could also be produced quickly in the meantime through the sea spray caused by the high wind speed.

19. Line 189: "…SSAs are strongly associated with the wind speed (Fig. S3)…" It is not clear how this conclusion is draw from Fig. S3.

Response: We found that we already described it in197 "SSAs are generated from sea bubble-bursting, which is mainly determined by the wind speed.". This is common sense. Thus, we removed this sentence to avoid misunderstanding. There is not necessary to repeat this old story.

20. Line 195: It would be helpful to show MSA/nssSO4 ratio and discuss the biogenic versus anthropogenic sulfur emissions and chemical formation.

Response: Thank you for your suggestions. However, $MSA/nss-SO_4^{2-}$ ratio could be strongly impacted by the temperature (Bate et al., 1992) by affecting the DMS oxidation pathway ($MSA/nss-SO_4^{2-}$ (%)= -1.5 T (°C) +42.2). Our cruise started from the low latitude ocean to the high AO. The temperature changed significantly along the cruise track. Thus, we can not use this ratio to analyse the anthropogenic sulfur emission.

Bates, T. S., J. A. Calhoun and P. K. Quinn (1992). "Variations in the methanesulfonate to sulfate molar ratio in submicrometer marine aerosol particles over the South Pacific Ocean." Journal of geophysical research **97**(D9): 9859-9865.

21. Line 200: Grammar issue with "such as.

Response: It was revised as "For instance".

22. Line 213: Grammar issue with "because".

Response: We revised this sentence as follows in lines 230-232:

But the amine fraction (5.23%) in the SL region was higher than that (4.2%) in the LL region, it could be attributed to the higher biogenic activity in SL than LL (Fig. S6) and the amines mainly originated from marine surfaces.

23. Line 227: "…Nss-SO42- correlated well with MSA in the HL region, indicating that nss-SO42- is mainly derived from the DMS…" Please show the scatter plot to get more information of the correlation. It is not clear to see by eye.

Response: We do the correlations and added them behind the sentence in line 245.

Nss-SO$_4^{2-}$ was correlated well with MSA in the HL region ($R^2$=0.647, n=56, Figure S8),

[Figure]

Figure S8, The relationship between MSA and Nss-SO$_4^{2-}$ in HL regions (75° - 85°N).

Note that we remove all the negative Nss-SO$_4^{2-}$ values to do this correlation.

24. Line 230-231: "…MSA is generated through alternative routes, including gas phase reaction and reactive uptake on the exiting particles…" Please provide more discussion on the chemical formation of MSA in the atmosphere. MSA formation also takes place in clouds in the marine boundary layer. Would it be fewer clouds in the HL for MSA formation?

Response: We presented the MSA formation information in the introduction part. There is no need to repeat the knowledge. Our description indicated that MSA could be produced as gaseous MSA directly or on the exiting particle through DMS oxidation. MSA formation can also take place in clouds as the clouds were the large drops in the marine boundary layer. Clouds might

facilitate the uptake of MSA, but the detail should be tested in future study.

25. Line 232-234: Please explain why low temperature was in favor of MSA formation in the atmosphere. Also, please show the scatter plot and correlation coefficient of MSA versus temperature.

Response: the previous work already demonstrated that the low temperature can enhance the DMS oxidation pathway to form MSA (Arsene et al. 1999). And, as we presented above, the correlation between MSA and temperature is not a new finding. Thus, we would not like to spend more description on this topic.

Arsene, C., I. Barnes, and K. H. Becker, 1999: FT-IR product study of the photo-oxidation of dimethyl sulfide: Temperature and O2 partial pressure dependence. Atmos. Chem. Phys., 1, 5463– 5470, doi:10.1039/A907211J.

The correlations were not significant. We can see the slope direction in Figure 4 a. We marked the slopes to easily distinguish the results from different regions.

26. Line 235-236: Again, please show the scatter plot and correlation coefficient of MSA versus temperature. It is difficult to tell by eye.

Response: We presented the slopes in Figure 4 a. The direction of slopes clearly indicated the negative or positive correlations between MSA and temperature in distinct regions.

27: Line 242-244: "…However, we did not find an obvious relationship between the MSA concentration and RH in the HL region. The MSA levels changed little with the RH in this region (Fig. 4b), indicating that RH had little effect on the MSA formation in the HL region…" Please explain why.

Response: There are two main routes for the production of MSA. One is the homogeneous reaction of DMS which is mainly related to the solar radiation and radicals. The other one is the heterogeneous reaction of DMS which is strongly influenced by RH. The result of no relationship between MSA and RH indicated that the MSA production was mainly dominated by the homogeneous reaction of DMS in high AO.

27. Line 249-250: Isn't DMS flux a function of DMS levels and transfer velocity? This does not logically make sense. Do you mean "transfer velocity increased with wind speed"?

Response: We revised this sentence as follows in lines 268-271:

Although high wind speed could lead to a high DMS flux by increasing the transfer velocity, the atmospheric DMS levels were possibly limited due to the low and unchanged surface seawater DMS levels (< 0.5 nmol L$^{-1}$) in the HL regions (Zhang et al., 2021).

28. Line 265: What type of radicals? How do they affect MSA formation? More discussion is needed.

Response: We are sorry for the confusing description here. We cited our previous finding that the radicals will strongly impact the atmospheric DMS oxidation process when air DMS concentration was high. We revised the sentence as follows in lines 286-287.

Low DMS concentration was the critical factor for MSA formation, but radicals, such as OH, NO3, BrO and ClO, would become the key parameter of the DMS chemistry when the DMS concentration was high (Yan et al., 2020b).

30. Line 266-268: "…The observation results confirmed that the low MSA concentration was determined by the low DMS concentration in the high AO and further demonstrated that low contribution of DMS chemistry was determined by the low DMS emissions in this region..." Why not determined by the radicals?

Response: During the summer season of AO, there happens on polar days. The concentration of radicals was strongly influenced by solar radiation. Therefore, if the radicals were enough to oxidize DMS, the production of sulfur aerosol will be tightly related to the DMS emission. However, the DMS emission in high AO was very low, it was reasonable that the contribution of DMS chemistry to atmospheric aerosol would be low.

---

## Author Comment (AC2)

**Response to reviewer 2**

**General response:**

**Dear anonymous reviewer 2,**

Thank you for your useful comments on our manuscript "Low contributions of dimethyl sulfide (DMS) chemistry to atmospheric aerosols over the high Arctic Ocean". We have carefully revised the manuscript as per the comments. Revisions in the text are shown using red highlight. The responses to the reviewer's comments are marked in blue with the corresponding changes highlighted in red and presented in the following.

**Summary/recommendations**

The authors present gas-phase MSA and particulate MSA, DMA, SO4, etc. results from ship cruises taken in the northern Atlantic ocean. The results indicate that DMS chemistry may not greatly impact aerosol composition or concentrations in the study region. I think that it is a good look at the problem, but that there is still more research to be done before declaring this finding as fact. The authors did not collect gas-phase DMS, they were only out for a single season, and their study range was small. I think the paper should be published with more caveats and care to not over-state their findings. The paper was well cited and generally well written, but there are several sentence fragments throughout the paper. I have pointed out a few but I do encourage the authors to carefully go through the paper again.

**General comments**

Introduction: it would be helpful to define the different Arctic regions more rigorouslycurrently the authors refer to 'low', 'high', and 'coastal'. Please provide latitude ranges or similar.

Response: We revised the introduction as comments. They are all defined in the introduction. As a general comment, it's helpful to define 'uncommon' acronyms in each major section when they first come up. (For example, AO and PO).

Response: We revised the introduction as comments.

Methods: How did the authors quality control their datasets to remove ship emission influences (that is, emissions from the R/V Xuelong)? This is an important but missing detail.

Response: We did the calibration cruise at the beginning and the end of the cruise. We have already tested it in the lab or on the other cruise in the Southern Ocean (Yan et al., 2019; Yan et al., 2020 a, b) and in the Arctic Ocean (Yu et al., 2020;2021). We deployed a total suspended particulate (TSP) sampler at the top of the mast (20 m height) to minimize the impact of ship emission. Conductive silicon tubing with an inner diameter of 1.0 cm was used to connect to the instrument to avoid the sampling loss of particles. We also removed the data which was impacted by the ship emission.

Methods: there is no discussion of the size range of aerosol collected, which is relevant for climate and human health. Was there any differentiation by size?

Response: Unfortunately, we did not measure particles in a different size. We only focused on the mass concentration of the total suspended particle.

Results: Since there was no direct measurement of DMS in the atmosphere, the findings are somewhat speculatory. The authors should state this limitation and be careful to not definitively state anything. The authors write as if they have solved this particular question for all of the AO (that DMS emissions and chemistry "scarcely impacted the atmospheric aerosols in the high AO" but I don't think this is so definitively solved. This study is a good look but requires more measurements/modeling over more seasons and a much larger area than the ship tracks covered. These caveats need to be discussed.

**Response:**

It is appreciated that the reviewer pointed out these weaknesses. However, it is very difficult for us to perform a perfect observation during the Arctic Ocean campaign, similar to a typical land station surrounding the Arctic Ocean. We can only obtain a short period of data across the central Arctic during our observation. We presented the details as follows to prove our conclusion of "low contributions of DMS chemistry to atmospheric aerosols over the high Arctic Ocean" was reliable and solid. Firstly, our previous study reported that deficient DMS levels ( $<0.5 \text{ nmol } L^{-1}$ ) and flux (general below 0.5 µmol m-2 d-1) were observed in the high latitude (Zhang et al., 2021, in Global Biogeochemical Cycles) due to the nutrient limitation and heavy sea ice cover. Even after sea ice retreat, the DMS levels remained unchanged, and the DMS flux slightly increased to only 1.2 µmol m-2 d-1. For the annual changes of DMS levels and flux, we can also conclude that these values were difficult to be changed as the upper layer water mass of Arctic Central was dominated by an increasing fresh water with very low nutrients (Figure 1, Zhang et al., 2021). Although we did not detect the atmospheric DMS and flux in the 2018 cruise, the low DMS flux and air DMS levels could be expected during the 2018 campaign.

Figure 1 Distribution of nutrients in relation to surface seawater dimethylsufide (DMS) in the western Arctic Ocean. (a) Locations of conductivitytemperature-depth (CTD) stations along T1. The section R, panels (c)–(f), correspond to stations labeled R on the map in (a). (b).

Relationships between DMS and Si and total N in surface water of all stations in (a). The panels on the right show depth profiles of (c) salinity, (d) Si, (e) total N, and (f) fluorescence along section R. (Zhang et al., 2021)

Secondly, we found that the MSA mass concentrations decreased from the low latitude Arctic to the high latitude. If the air mass contained high MSA levels could be rapidly transported to high latitude regions, we would observe high MSA in the high Arctic. However, we found deficient gas phase and particle MSA levels over there. In addition, MSA is well known only from DMS oxidation in the atmosphere. This means that the oceanic DMS emission greatly

influences the atmospheric MSA production. Thus, the low emission of DMS in the high Arctic was possibly the main reason for extremely low MSA there.

Thirdly, our observation of aerosol compounds indicated that the contributions of biogenic sulfur decreased significantly (only 1.61%) in high latitude Arctic (Figure 4 in manuscript). This result suggested that the low DMS chemistry contribution to aerosol was found in the high Arctic.

Lines 107-109: provide latitude ranges for LL, ML, SL, and HL. These ranges appear to be in Table S3 but I think this is relevant enough to be in the main text as well. Response: We agreed that and Table S3 was moved to the main text.

Lines 113-114: "The variations in the MSAg level were not always consistent with those of the MSAp along the cruise tracks, indicating that the formation mechanisms of MSAg and MSAp from oxidation of DMS are different." This is an interesting statement, I think the authors could expand upon it some. Consider providing a supplemental figure or table showing the ratio of MSAg / MSAp. This may be of research interest to some.

Response: Thanks for the reviewer giving this good suggestion. Actually, we are preparing another manuscript to focus on this topic. And we want to compare the difference in this ratio between the Arctic Ocean and the Southern Ocean. Additionally, we have already published one paper about this topic: observation in the Southern Ocean (Yan et al., 2019 c).

Line 119-120: "The variations in MSAp levels in the SL region during leg I and II were associated with the phytoplankton activity in these regions (Fig. S6)." It is not easy to make this association based on looking at Fig S6 and Fig 1. The authors need to do more work here to make this association more clear. It would be nice to have some sort of statistical measure of how well associated MSAp and phytoplankton are - this could be an average for the region or more high-definition. The authors may also consider adding a supplemental figure that shows this association more clearly, like the ratio of phytoplankton to MSAp along the ship tracks, if possible.

Response: Thanks for the reviewer giving this good suggestion. We did not make the

correlations between the Chl a and MSAp in distinct regions. Our description is mainly based on the difference of Chl a, an indicator of phytoplankton activity, values in a large-scale scan. As in figure S6, we could easily find the difference of Chl a in the distinct regions. This difference was consistent with the variations in MSAp along the cruise track. Our purpose here was to note that the high MSA occurred in the regions where phytoplankton activity was high. Line 209-210: I recommend that the authors state "That is likely the reason why..." as the authors did not do a comprehensive analysis (for example, a principal components analysis) on where emissions were coming from for each species.

Response: Sure, we agreed with this comment, and the sentence was revised in line 230.

Line 212: do the authors have the measurement precision to state that a  $\sim 1\%$  difference in amine is significant?

Response: Our measurement precision for amine is 0.12 ng m-3. We did not consider that the 1% difference in amine in SL and LL regions was significant. We just presented the difference but no more description or explanation for this small difference. And, we revised these sentences in lines 230-233 as follows:

Similar to  $NO_x^-$ , high fractions of amine aerosols were found in the SL and LL regions. But the amine fraction (5.23%) in the SL region was higher than that (4.2%) in the LL region because of amine emissions from marine surfaces.

Technical comments

Line 22: HL not defined in the abstract

Response: It was revised in line 27. "high latitude (HL) region"

I recommend against using undefined acronyms in your short summary.

Response: It was revised as commented. We did not use acronyms in the short summary.

Line 29: "The Arctic is known for its amplified rate? of global climate change..." there is a

missing word, perhaps the authors meant rate? Or Amplification instead of amplified?

Response: We replace the amplified with "amplification".

Line 34: no comma needed between "increase, when"

Response: We deleted the comma.

Line 48 - I recommend adding a citation for "The loss of sea ice in the AO promotes the air-sea

exchanges and subsequently increases dimethyl sulfide (DMS) emissions." - unless the Sharma et al 2012 citation was appropriate for this statement as well? This isn't clear to me. Response: Sure, we added a citation (Galí et al., 2019) here in line 54.

Galí, M., E. Devred, M. Babin and M. Levasseur (2019). "Decadal increase in Arctic dimethylsulfide emission." Proceedings of the National Academy of Sciences 116(39): 19311-19317.

Line 53: "The observations carried out in the Ny-Ålesund revealed..." Do the authors mean Ny-Alesund region?

Response: Yes, Ny-Ålesund region.

Lines 98-99: the sentence here is a sentence fragment.

Response: We revised the sentence as follows in lines 109-110.

The remote sensing chlorophyll-a data was obtained from MODIS-Aqua (http://oceancolor.gsfc.nasa.gov) with a spatial resolution of 4 km.

Results: the authors are inconsistent in whether or not they capitalize 'leg'.

Response: We checked the whole manuscript. If the leg started the sentence, we used the capitalized "Leg".

Line 116: I think the authors meant "confirming" not "conforming"?

Response: Thanks. We revised the sentence.

Line 135: the sentence here is a sentence fragment, I believe it belonged to the previous sentence? Response: We revised all these sentences from line 152 to line 156 as follows:

Although obvious sea ice retreat occurred from July to September (Fig. S7), the chlorophylla remained at an extremely low level in this region possibly leading to very low DMS emission (Zhang et al., 2021, Figure S6). This would be the main reason for low observed sulfur aerosols. In contrast, our result differed from the observation in the high latitude SO that high atmospheric DMS chemistry contribution was reported when the sea ice retreated (Yan et al., 2020 a). Lines 167-168: the sentence here is a sentence fragment, I believe it belonged to the previous sentence?

Response: We revised the sentence in lines 187-189 as follows:

Because Na+ and MSA can be used as a marker for sea salt aerosols and biogenic sulfur aerosols, respectively, the variations in the MSA to Na+ ratio is useful to understand the contribution of biogenic sulfur species in the marine atmospheric aerosols.

Line 213: sentence fragment.

Response: It was revised in lines 231-233.

But the amine fraction (5.23%) in the SL region was higher than that (4.2%) in the LL region, which could be attributed to the higher biogenic activity in SL than LL (Fig. S6), and the amines mainly originated from marine surfaces.

Line 258: suggest "strong positive..." rather than good. This is a more common way of phrasing it. (E.g. strong, moderate, weak correlations)

Response: It was revised as comments in line 281.

Line 277: Suggest "Sulfuric acid is more effective for new particle formation"

Response: It was revised as comments in line 300.

Line 305: The authors may have meant "concerning", not "concerned".

Response: It was revised as comments. We used "be highly paid attention" here.

Citation for "Croft, B., Martin, R. V., Leaitch, W. R., Tunved, P., Breider, T. J., D'Andrea, S. D., and Pierce, J. R.: Processes controlling the annual cycle of Arctic aerosol number and size distributions, Atmos. Chem. Phys., 16, 3665–3682, doi:10.5194/acp-16-3665-2016, 2016. " - fix the D'Andrea, and this is erroneously "Crof et al 2016" in the main text, line 59. Fix to Croft.

Response: Sure. It was revised as comments.

**Figures/Tables**

All figure with spatial maps (e.g. Fig 1): I recommend considering explicitly drawing out the different characterized regions (ML, SL, HL, LL). For example, you could make the land/sea masses a lighter gray, then add dashed lines to separate each region + a label on the figure for

each region. This would make the results more distinct.